# Learning Relation Entailment
# with Structured and Textual Information

**Zhengbao Jiang**[1]**, Jun Araki**[2]**, Donghan Yu**[1]**, Ruohong Zhang**[1]**,**
**Wei Xu**[3]**, Yiming Yang**[1]**, Graham Neubig**[1]

[1] *Language Technologies Institute, Carnegie Mellon University*
{ZHENGBAJ, DYU2, RUOHONGZ, YIMING, GNEUBIG}@CS.CMU.EDU

[2] *Bosch Research North America*
JUN.ARAKI@US.BOSCH.COM

[3] *Department of Computer Science and Engineering, Ohio State University*
XU.1265@OSU.EDU

## Abstract

Relations among words and entities are important for semantic understanding of text, but previous work has largely not considered *relations between relations*, or *meta-relations*. In this paper, we specifically examine *relation entailment*, where the existence of one relation can entail the existence of another relation. Relation entailment allows us to construct relation hierarchies, enabling applications in representation learning, question answering, relation extraction, and summarization. To this end, we formally define the new task of predicting relation entailment and construct a dataset by expanding the existing Wikidata relation hierarchy without expensive human intervention. We propose several methods that incorporate both structured and textual information to represent relations for this task. Experiments and analysis demonstrate that this task is challenging, and we provide insights into task characteristics that may form a basis for future work. The dataset and code have been released at https://github.com/jzbjyb/RelEnt.

## 1 Introduction

Relations among words or entities play a fundamental role in semantic understanding of text, to the point where the dictionary definition of lexical semantics explicitly references "semantic relations that occur within the vocabulary" [Geeraerts, 2017]. Because of this, there are many curated relational knowledge bases (e.g. Wikidata, DBpedia, Freebase, OpenCyc, YAGO [Färber et al., 2015]), and many works have examined automatic extraction of relations from raw text [Surdeanu and Ji, 2014, Chaganty et al., 2017] or use of relations in applications such as question answering [Cui et al., 2017]. Typically, relations are treated as independent; for example, relation extraction and knowledge base completion are usually formulated as multi-class or multi-label classification problems [Hendrickx et al., 2010, Riedel et al., 2010, Bordes et al., 2013], where each relation is treated as an independent class with separate parameters.

We argue that this isolated view of relations is too simplistic, and that there are in fact *relations between relations*, or *meta-relations* that are interesting to study from both philosophical and practical perspectives. We specifically examine *relation entailment* in this paper, where the existence of one relation can entail the existence of another relation. This allows us to organize relations into an underlying hierarchy, with more abstract relations at the top of the hierarchy and more specific relations at the bottom; child relations entail their parent relations. An example for the creator relation is shown in Figure 1. creator is a generic relation that can describe the relation between

books and writers, paintings and painters, or software and programmers, while `developer` can only be used for the last case, thus being more specific.

Some previous works have inferred entailment among *textual relational patterns*, e.g. "⟨singer⟩ sings ⟨song⟩" is semantically subsumed by "⟨musician⟩ performs on ⟨musical composition⟩" [Lin and Pantel, 2001, Nakashole et al., 2012, Grycner et al., 2015, Kloetzer et al., 2015]. In contrast, we focus on determining entailment among *canonicalized relations* in a knowledge graph (KG), e.g. `developer` entails `creator`, where each relation is associated with entities. We are particularly interested in this setting because if these hierarchies can be robustly learned for relations, they could be used in a wide variety of downstream applications related to KGs, such as knowledge graph representation learning, KG-based question answering (QA) systems, and relation extraction. In fact, manually-created relation hierarchies have already been applied to a subset of these applications, improving consistency of distant supervision for relation extraction [Han and Sun, 2016], or consistency of learned graph representations [Zhang et al., 2018]. Relation entailment also has the potential to incorporate

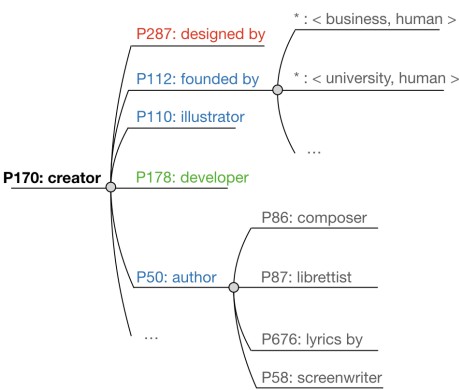

Figure 1: An excerpt of the Wikidata relation hierarchy. First tier relations are split into train/dev/test sets and `creator` is their parent. Other tiers are split similarly. * denotes pseudorelations split from leaf relations.

entirely new relations into a relation typology, expanding the knowledge graph to new relation types or entirely new domains.

In this paper, we introduce a new task of predicting relation entailment: given two relations predicting whether one entails the other. We first build a **Rel**ation **Ent**ailment dataset (**RelEnt**) by expanding the existing Wikidata relation hierarchy, including 3,551 relations with entailment annotation. We propose methods to represent the semantics of each relation using entities connected by the relation, and/or textual context. We use these methods to establish several baselines for this task. Empirical analysis demonstrates that predicting relation entailment is a challenging problem that requires high-level abstraction, and given that existings datasets are relatively small, it provides a unique testbed for evaluating models' generalization ability from sparse learning signals.

## 2  Definition of Relation Entailment

For simplicity of discussion, we consider binary relations that connect a head entity (subject) $h$ to a tail entity (object) $t$. Binary relations are used in the great majority of work on relation extraction and knowledge base completion [Surdeanu and Ji, 2014]. The following definitions and formulations can also be extended to $n$-ary relations.

Let $\mathcal{E}$ denote the set of entities and $\mathcal{R}$ denote the set of relations. A knowledge base is a collection of facts represented in triplets $\mathcal{C} = \{\langle h^{(i)}, r^{(i)}, t^{(i)} \rangle\}_{i=1}^{|\mathcal{C}|}$, where $h, t \in \mathcal{E}$, $r \in \mathcal{R}$, and a triplet $\langle h, r, t \rangle$ indicates that $h$ and $t$ have a specific relation $r$. For definition purposes, we assume the ideal setting of the knowledge base $\mathcal{C}$ including *all* valid triplets, which conversely means that if $\langle h, r, t \rangle \notin \mathcal{C}$, $h$ and $t$ do not have relationship $r$. As we will show later, in a more realistic setting, knowledge bases are inherently incomplete and the entailment of relations can only be inferred from noisy facts $\hat{\mathcal{C}}$, instead of directly from $\mathcal{C}$.

| Original relations | Pseudo-relations after splitting |
|---|---|
| `parent_organization` | ⟨laboratory, university⟩, ⟨airline, airline⟩, ⟨record label, record label⟩, ... |
| `architectural_style` | ⟨railway station, architectural style⟩, ⟨church, architectural style⟩, ... |
| `award_received` | ⟨film, Academy Awards⟩, ⟨human, campaign medal⟩, ⟨human, scholarship⟩, ... |

Table 1: Pseudo-relations from type-based splitting. Each tuple is the types of head/tail entities.

**Definition 1.** Let $\mathcal{C}_r$ denote the set of all head-tail entity pairs $\langle h, t \rangle$ that have a specific relation $r$, that is: $\mathcal{C}_r = \{\langle h, t \rangle | \langle h, r_i, t \rangle \in \mathcal{C}, r_i = r\}$. Each $\langle h, t \rangle$ is considered an *instance* of relation $r$.

**Definition 2.** A relation $r$ entails $r'$, denoted as $r \models r'$, if and only if $\mathcal{C}_r$ is contained by $\mathcal{C}_{r'}$: $r \models r' \Leftrightarrow \mathcal{C}_r \subseteq \mathcal{C}_{r'}$. We call $r$ the *child relation* of its *parent relation $r'$*.

Intuitively, $r$ can be viewed as a special case of $r'$ if $r \models r'$, for example, `author` $\models$ `creator`. The `author` of a book, screenplay, or piece of music is also considered as its `creator`. By definition, entailment is transitive: $r \models r'$ and $r' \models r'' \Rightarrow r \models r''$. This entailment implies a hierarchical structure among relations, e.g. in Figure 1 `creator` is the parent relation of `author`, `illustrator`, etc. The higher the relation is in the hierarchy the more abstract.

In this paper, we study the problem of predicting relation entailment. Unlike the definition above, where we considered an idealized complete knowledge base that provides all instances of each relation, real-world knowledge bases will never cover all existing relations for all entities. Thus, we have only an incomplete and noisy version of $\mathcal{C}_r$ and $\mathcal{C}_{r'}$, denoted as $\hat{\mathcal{C}}_r$ and $\hat{\mathcal{C}}_{r'}$, respectively. Because of this, $r \models r' \not\Leftrightarrow \hat{\mathcal{C}}_r \subseteq \hat{\mathcal{C}}_{r'}$, and we can not use the definition above to determine whether entailment exists. Formally, we frame the task of predicting relation entailment as follow:

**Task Definition** Let $\mathcal{L}$ denote a set of parent relations, given a set of $N$ training child relations $\mathcal{X}_{\text{train}} = \{r_i\}_{i=1}^{N}$ and their ground truth parents $\mathcal{Y}_{\text{train}} = \{r_i'\}_{i=1}^{N}$ (i.e., $r_i \models r_i', r_i' \in \mathcal{L}, i = 1, \ldots, N$), we want to predict the parents for $M$ relations in the test set $\mathcal{X}_{\text{test}} = \{r_j\}_{j=N+1}^{N+M}$.

This task can be conceptually interpreted as a multi-class classification problem, where parent relations $r' \in \mathcal{L}$ are classes and we want to assign each child relation the most specific relation (class) it entails, which would become its immediate parent in a relation hierarchy. The only difference is that child relation $r$ and parent relation $r'$ can be represented similarly by sets of instances $\hat{\mathcal{C}}_r$ and $\hat{\mathcal{C}}_{r'}$. The goal is to generalize from the existing (training) child relations in the knowledge base to new unseen (test) child relations, determining which parent relation it belongs to, e.g., given that `author` is a child relation of `creator`, can we predict that `developer` also entails `creator`?

## 3 RelEnt Dataset

As training and testing data for relation entailment prediction, we construct a **Rel**ation **Ent**ailment dataset (**RelEnt**) from Wikidata, a widely used large scale knowledge graph.[1] Wikidata contains relations that have been manually curated and organized as a hierarchy as shown in Figure 1, where each relation has no or one parent relation.[2] After removing relations whose heads or tails are not entities (e.g. the tails of `image` relation are actual pictures), there are 1,240 relations, among which 296 have parents. We perform the following steps to create the RelEnt dataset:

---

1. We use 2018-09-21 version from https://dumps.wikimedia.org/. Relations are called "properties" in Wikidata.
2. 17 out of 296 relations have multiple parents. Here, to avoid ambiguity we randomly keep one of them and remove the others, and leave dealing with multiple entailment to future work.

1) **Instance Set Creation** We create instance set $\hat{\mathcal{C}}_r$ for each relation by collecting all the head-tail entity pairs in the KG that have a specific relation $r$. Because both parent and child relations come from the same data source (Wikidata), there are some cases where many of the instances of the child relation are also manually annotated as instances of the parent relation, which makes the problem of predicting entailment relatively easy. However, when trying to reason over relations in disparate knowledge bases or predict entailment for relations that were automatically extracted from text, this level of overlap cannot be expected. Thus, to ensure that our dataset is sufficiently representative of these more difficult cases, we ensure that the parent relation $r'$ has *no* instance overlap with the children relation $r$ when predicting its parent.

2) **Downsampling** Since the original Wikidata is unwieldy to experiment on in its entirety, we follow Bordes et al. [2013] and downsample the instances. To keep as many relations as possible and avoid removing relations with few head-tail pairs, the downsampling is conducted on a per-relation basis. We keep $10 \times \sqrt{\min(|\hat{\mathcal{C}}_r|, 10^5)}$ instances for each relation, where $|\hat{\mathcal{C}}_r|$ is the total number of instances. Like Bordes et al. [2013], we keep the top instances according to average frequency in the Wikipedia corpus of the head and tail entities, which helps ensure sufficient textual information for training the text-based representations.

3) **Relation Expansion** Since the number of relations with manually created parent annotations in Wikidata is small, the supervision provided is sparse. To increase the number of relations that can be used for learning, we expand the relation hierarchy by splitting leaf relations (i.e. relations without children) into multiple pseudo-relations based on the type of the head entity and tail entity. For example, relation `founded_by` in Figure 1 is split into multiple pseudo-relations, where `business` and `university` are the types of the head entities and `human` is the type of the tail entities. Although these pseudo-relations do not correspond to relations explicitly labeled in Wikidata, because relations are usually associated with certain types of head and tail entities [Jain et al., 2018] it is arguably reasonable to treat these relations as distinct (some examples in Table 1). This also greatly increases the number of relations for training and testing without human annotation effort. Since the original leaf relation disappears after splitting, we randomly choose one of the pseudo-relations to fill the vacancy and all the others serve as its children.[3]

4) **Entity Linking** To extract textual contexts for each relation, we run SLING,[4] a frame semantic parser [Ringgaard et al., 2017], to identify mentions of Wikidata entities in the Wikipedia corpus.

5) **Train/Dev/Test Split** We divide all the relations into train/dev/test sets by splitting each tier of the hierarchy randomly and using their parents as labels. The example in Figure 1 shows how the first tier of the hierarchy is split into the train/dev/test sets with `creator` as the parent, and the same procedure applies to other tiers. Like Chen et al. [2019], we remove relations with less than 10 instances from the splits and leave dealing with few-shot relation entailment to future work. The statistics of the RelEnt dataset are listed in Table 2, where 86% of parent relations have more than one child.

---

3. We propagate instances from child relations to parent relations so that the parent pseudo-relation contains other pseudo-relations placed below it. As a result, pseudo-relations do not violate Definition (2). At training time, when predicting the parent of a particular relation $r$, we will first remove $r$'s instances from its parent to avoid making this learning problem trivial.

4. https://github.com/google/sling

| Statistics of the downsampled Wikidata | | | Number of relations in each split | | | |
|---|---|---|---|---|---|---|
| #Relations | #Entities | #Triplets | Train | Dev. | Test | Classes |
| 15,658 | 434,654 | 3,031,176 | 2,055 | 804 | 692 | 498 |

Table 2: RelEnt dataset statistics. Each class is a parent relation. Each triplet is a fact $\langle h, r, t \rangle$.

| Method | TransE | DistMult | ComplEx |
|---|---|---|---|
| **Score** | $-\|\mathbf{h} + \mathbf{r} - \mathbf{t}\|_{\frac{1}{2}}$ | $\mathbf{h}^\top \mathrm{diag}(\mathbf{r})\mathbf{t}$ | $\mathrm{Re}(\mathbf{h}^\top \mathrm{diag}(\mathbf{r})\bar{\mathbf{t}})$ |

Table 3: Knowledge graph embedding methods and their score functions. $\mathbf{h}$, $\mathbf{r}$, and $\mathbf{t}$ are embeddings of the head, tail, and relation respectively. $\bar{\mathbf{t}}$ is the conjugate of $\mathbf{t}$. $\mathrm{Re}(\cdot)$ means taking the real part of a complex value [Wang et al., 2017].

## 4  Relation Representation

Now that we have created a dataset, we describe models for predicting relation entailment. We define a score function $s(r, r')$ to evaluate how likely it is that $r$ entails $r'$. This scoring function takes in a representation of each relation, and in this section we discuss three overall methods for representations: embeddings based on structured information from the KG, embeddings based on textual information, and distribution-based representations. We will describe how to convert these representations into scores in the following section.

### 4.1  Embedding with Structured Information

The first way we examine representing relations is through knowledge graph embeddings (KGE) either representing the relation $r$ itself, or representing the instances $\hat{\mathcal{C}}_r$ existing in the knowledge base. The basic idea of KGE is that they are trained to assign higher scores to positive triplets in $\hat{\mathcal{C}}$ than negative triplets, learning embeddings for entities and relations that are consistent with the knowledge base data. We use three KGE methods in our experiments, namely TransE [Bordes et al., 2013], a method based on additive relational embeddings, DistMult [Yang et al., 2015], a method based on multiplicative relational embeddings, and ComplEx [Trouillon et al., 2016], a method based on complex number relational embeddings to model asymmetric relations. Based on these embeddings, we examine two ways to represent a relation as a vector $\mathbf{e}_r$.

**Relation Embedding**  Directly use relation embeddings generated by KGE methods: $\mathbf{e}_r = \mathbf{r}$. Similar relations tend to be close to each other in the embedding space, so we hypothesize that the information contained in relation embeddings could also be used in entailment prediction.

**Head-tail Entity Aggregation**  Given multiple head-tail entity pairs of a relation, we can also assume that they will be indicative of the relation. For example, for `author` most participating heads and tails will be books and writers, while `developer` will have software and programmers. We attempted several methods to aggregate these head and tail embeddings into a single one for a relation, but found that concatenating the head and tail (represented as $[\cdot ; \cdot]$) then mean pooling all instances to be effective: $\mathbf{e}_r = \frac{1}{|\hat{\mathcal{C}}_r|} \sum_{\langle h, t \rangle \in \hat{\mathcal{C}}_r} [\mathbf{h}; \mathbf{t}]$.[5]

---

5. We also tried graph neural networks (GNN) [Li et al., 2016, Kipf and Welling, 2017], which propagate representations of neighbouring nodes to head and tail entities. These methods did not show significant improvements; we conjecture this is because KGE is already trained using the graph, and most potential gains have already been achieved.

## 4.2 Embedding with Textual Information

While the above structured methods are a reasonable start, in many cases simply using head and tail entities is not sufficient to represent the meaning of relations. For example, both `film_editor` and `director` are used to connect a movie to crew members. Thus their head and tail entity embeddings are very close and hard to distinguish. As a consequence, child relations of `film_editor` might mistakenly be predicted to have `director` as their parent. To address this problem, we use mentions of head and tail entities in Wikipedia to extract textual context to enrich the relation representations.[6] Given the fact that `film_editor` usually co-occurs with words like "edit" and `director` usually co-occurs with words like "direct", they can be better distinguished from each other with textual information.

Using distant supervision [Mintz et al., 2009], we first extract all sentences that contain both head and tail entities for each relation $r$ from Wikipedia sentences $\mathcal{S}_r = \{s|h \in e(s) \wedge t \in e(s), \exists \langle h, t \rangle \in \mathcal{C}_r\}$, where $e(s)$ includes all entities detected from sentence $s$ in the entity linking step detailed above. Given the bag of sentences $\mathcal{S}_r$, we consider the following two ways to extract textual context. **(1) Words in the Middle**: based on the observation that words in the middle of the head and tail entity often describe their relation, we use those words to represent the meaning of the relation: $\mathcal{U}_r^{\text{mid}} = \{\text{mid}(s)|s \in \mathcal{S}_r\}$, where $\text{mid}(\cdot)$ returns the words in the middle. **(2) Dependency Path**: Toutanova et al. [2015] have noted that it can be more accurate to capture relations through dependency paths between head and tail entities, as shown in Table 5: $\mathcal{U}_r^{\text{dep}} = \{\text{dep}(s)|s \in \mathcal{S}_r\}$, where $\text{dep}(\cdot)$ returns the lexicalized dependency path as a sequence of tokens.

Given extracted textual contexts $\mathcal{U}_r$ (mid or dep), we model them either as a bag-of-tokens (BOT), or consider them as sequences of tokens. Both words and dependency arcs are treated as tokens and represented with embeddings.

**BOT Representation** The simplest way to use textual context $\mathcal{U}_r$ is to model all the tokens as a bag-of-tokens: $\mathbf{e}_r = \frac{\sum_{v \in \mathcal{V}} n_v^r \cdot \mathbf{w}_v}{\sum_{v \in \mathcal{V}} n_v^r}$, where $v$ is a token, $\mathcal{V}$ is the vocabulary, $\mathbf{w}_v$ is the embedding of the token, $n_v^r$ is the frequency of the token in all the contexts of the relation $r$. To avoid overfitting, we only use the most frequent $k$ tokens after removing stopwords.

**Sequential Representation** To further consider the order of the $n$-token context $u$, we use BiL-STMs [Hochreiter and Schmidhuber, 1997] or CNNs [Kim, 2014] to extract contextual embeddings: $\mathbf{e}_u = \text{BiLSTM}(\mathbf{w}_1, \mathbf{w}_2, ..., \mathbf{w}_n)$ or $\mathbf{e}_u = \text{CNN}(\mathbf{w}_1, \mathbf{w}_2, ..., \mathbf{w}_n)$, where BiLSTM concatenates the hidden representation of the first and last position, and CNN uses mean pooling. We also tried the embedding average as a simple sequence representation: $\mathbf{e}_u = \text{mean}([\mathbf{w}_1, \mathbf{w}_2, ..., \mathbf{w}_n])$. Given the representation of each context, the representation of a relation is the average of context representations weighed by their counts: $\mathbf{e}_r = \frac{\sum_{u \in \mathcal{U}_r} n_u \cdot \mathbf{e}_u}{\sum_{u \in \mathcal{U}_r} n_u}$, where $n_u$ is the count of the context $u$. To limit computational cost and avoid overfitting, we only use the most frequent $k$ contexts in $\mathcal{U}_r$.

## 4.3 Distribution-based Representations

All previous methods attempt to aggregate all instances into a single vector. We hypothesize that the instances of the child relation occupy a subspace of the parent relation, so the overall volume of the child will be smaller and contained within the parent, which can not be captured by a single vector as discussed in [Vilnis et al., 2018]. To this end, we model instances as a distribution over embedding space. While there are many ways to representations, here we use kernel density estimation (KDE)

---

6. We assume no access to the names of the relations because the pseudo-relations do not have names.

with a Gaussian kernel $K$ and width $w$ to represent the distribution, which is a non-parametric method to estimate a distribution based on samples drawn from it: $P^r(\mathbf{e}) = \sum_{\langle h,t \rangle \in \hat{\mathcal{C}}_r} K(\frac{\mathbf{e}-[\mathbf{h};\mathbf{t}]}{w})$.

## 5  Entailment Prediction

Given the representations of relations $\mathbf{e}_r$, we use a score function $s(r, r')$ to evaluate how likely it is that $r$ entails $r'$. Candidate parents are ranked in descending order and the one with highest score is chosen as parent. To better understand how entailment is different from similarity measurement, we start with unsupervised similarity measurements, then introduce learnable score functions.

### 5.1  Unsupervised Scoring Functions

Specifically, we examine three varieties of unsupervised scoring functions. **(1) Cosine Similarity**: $s(r, r') = \cos(\mathbf{e}_r, \mathbf{e}_{r'})$. **(2) Euclidean Similarity**: $s(r, r') = -\text{euc}(\mathbf{e}_r, \mathbf{e}_{r'})$. **(3) KL Divergence**: when using a distribution-based representation, we use KL-divergence to measure the similarity between the distribution of child relation ($P^r$) and parent relation ($P^{r'}$): $s(r, r') = -D_{\text{KL}}(P^r || P^{r'})$. An advantage of using KL-divergence is that it is asymmetric: the distribution of parent relation ($P^{r'}$) needs to cover the distribution of child relation ($P^r$) to have high score, which is desirable in the case of inferring asymmetric entailment.

When calculating these unsupervised metrics, we also optionally perform *relation instance propagation*. The ability of these metrics to generalize is based on the assumption that some child relations of a parent in the training set might be similar to their siblings in the test set. However, the above methods do not use any supervision and the parent relation is not aware of its children in the training set, thus limiting its coverage. To overcome this issue, we propagate instances from children in the training set up into their parents, i.e., $\hat{C}_{r'} = \bigcup_{r \in \text{child\_in\_train}(r')} \hat{C}_r \cup \hat{C}_{r'}$, and we hypothesize that after propagation, parent relations become closer to their children in the test set.

### 5.2  Supervised Scoring Functions

The above score functions are fixed similarity measurements without learnable parameters, limiting their modeling power. In the supervised setting, we use two multilayer perceptron (MLP) to project representations of child and parent relation into another space where entailment is measured by dot product: $s(r, r') = \text{MLP}_c(\mathbf{e}_r) \cdot \text{MLP}_p(\mathbf{e}_{r'})$.[7] We use softmax to get an entailment probability distribution over all parents and optimize with cross-entropy loss.

## 6  Experiments

To (1) evaluate the difficulty of this new task of predicting relation entailment, and (2) compare and contrast each method described above, we conduct a series of experiments using the RelEnt dataset.

**Evaluation Metrics**  Since all the candidate parent relations are ranked by their scores, we compute several ranking metrics to measure the performance based on the rank position of the correct parent, including accuracy@1 (**Acc@1**), accuracy@3 (**Acc@3**), and mean reciprocal rank (**MRR**).

**Baselines**  We experiment with both unsupervised and supervised methods. For unsupervised methods, we only use representations from KGE methods since textual information involves learnable parameters in the BiLSTM and CNN. **Relation** refers to relation embeddings from KGE methods.

---

7. Propagation is done similarly in supervised settings, except that during training with a child relation $r$, we exclude its instances from the populated parent to avoid making it trivial.

**HT dist** models head-tail entitiy embeddings as a distribution and uses KL-divergence to compute similarity. **HT euc** aggregates head-tail entity embeddings and uses Euclidean distance to compute similarity, while **HT cos** uses cosine to compute similarity. Since the best performing unsupervised method is head-tail entity embedding aggregation with TransE, we train a MLP based on this representation, which serves as the **base model** for other supervised models. We extend this representation by concatenating embeddings from textual information. **BOT** uses all the tokens from contexts as a bag-of-tokens. **Avg** represents each context with embedding average. **CNN** represents each context with CNN, while **BiLSTM** uses BiLSTM.

**Implementation Details** We sample 100 instances for each relation. We use spaCy[8] to extract dependencies between head and tail entities. To mitigate noise introduced by distant supervision, we only keep contexts that occur more than 10 times for each relation. We use the most frequent $k = 10$ tokens/contexts in bag-of-token/sequential representations. We use implementations of TransE, DistMult, and ComplEx in PyTorch big graph [Lerer et al., 2019] to train 200-dimensional embeddings for 50 epochs. Gaussian kernel width is set to 0.1. We use 50-dimensional GloVe embeddings [Pennington et al., 2014], with out-of-vocabulary words and dependency arcs initialized randomly, which are updated during training. We use a single-layer BiLSTM with 64 hidden states, and a CNN with window size of 3 and 64 filters. Two two-layer MLPs with 256 hidden states and dropout [Srivastava et al., 2014] of 0.5 are used for child and parent relations respectively.

### 6.1 Experimental Results

**Overall Performance** The performances of unsupervised and supervised methods are listed in Table 4a and Table 4b. Overall, supervised methods with learnable parameters perform significantly better than unsupervised methods. The best-performing unsupervised method (**HT cos** with TransE and propagation) achieves an accuracy of 0.572, which means that for about half of the test relations, their parents are simply the most similar ones from the candidates. This is expected because a relation tends to be closer to its parent than non-parents. However, embeddings derived from KGE methods are not calibrated for entailment prediction. As we will analyze later, through projecting the embeddings to another space, the accuracy improves to 0.681 (**base model**).

**Unsupervised Methods** As shown in Table 4a, we found that (1) TransE, despite its simplicity, performs better than DistMult and ComplEx across all similarity measurements. We conjecture that this is because the additive nature of TransE makes it better for entailment prediction than the multiplicative calculation used in DistMult and ComplEx, especially when we directly use relation embeddings derived from these methods. (2) Head-tail embeddings yield better performance than relation embeddings, perhaps because head-tail embeddings are more expressive, which is consistent with the observations in Chen et al. [2019]. (3) Aggregation (**HT euc** and **HT cos**) performs better than distribution (**HT dist**). We argue that distribution-based methods are more prone to noise than aggregation-based methods, because density estimation is more likely to be affected by a single instance than a simple average. (4) Propagation improves the performance substantially. Since parent relations have no overlap with child relations initially, populating parent relations broadens their coverage and enriches their representations.

**Supervised Methods** As shown in Table 4b, we can conclude that (1) Textual information is complementary to structured information. Both words in the middle and dependency paths improve the performance significantly, indicating that textual contexts provide additional signal for entailment

---

8. https://spacy.io/

| Prop. | Method | TransE | DistMult | ComplEx |
|---|---|---|---|---|
| - | Relation | .327 | .084 | .084 |
| ✗ | HT dist | .393 | .322 | .342 |
| | HT euc | .418 | .292 | .344 |
| | HT cos | .415 | .316 | .350 |
| ✓ | HT dist | .506 | .445 | .477 |
| | HT euc | .491 | .335 | .397 |
| | HT cos | **.572** | **.486** | **.501** |

(Table 4a) Accuracy@1 of unsupervised methods with different knowledge graph embeddings, similarity measurements, and whether to use relation instance propagation.

| Context | Method | Acc@1 | Acc@3 | MRR |
|---|---|---|---|---|
| | Base model | .681 | .863 | .779 |
| Words | + BOT | .698 | .855 | .785 |
| | + Avg | .701 | .857 | .786 |
| | + CNN | .705 | .858 | .790 |
| | + BiLSTM | .706 | .861 | .791 |
| Dep. | + BOT | .694 | .862 | .784 |
| | + Avg | .709 | .868 | .796 |
| | + CNN | .712 | .864 | .795 |
| | + BiLSTM | **.712** | **.872** | **.798** |

(Table 4b) Performance of supervised methods using both structured and textual information. All metrics are averaged across 5 runs with different random seeds. Note that for comparison the base model only uses structured information.

| Relation | Head and tail entities | Textual contexts |
|---|---|---|
| place_of_death | ⟨Captain Nemo, Pacific Ocean⟩ | $\xleftarrow{\text{nsubj}}$ died $\xrightarrow{\text{prep}}$ in $\xrightarrow{\text{pobj}}$ |
| place_of_birth | ⟨Julius Caesar, Rome⟩ | $\xleftarrow{\text{nsubjpass}}$ born $\xrightarrow{\text{prep}}$ in $\xrightarrow{\text{pobj}}$ |
| military_branch | ⟨Ronald Reagan, United States Army⟩ | $\xleftarrow{\text{pobj}}$ under $\xleftarrow{\text{prep}}$, $\xleftarrow{\text{poss}}$ divisions $\xrightarrow{\text{compound}}$ |
| commander_of | ⟨Joseph Stalin, State Defense Committee⟩ | $\xrightarrow{\text{appos}}$ commander $\xrightarrow{\text{prep}}$ of $\xrightarrow{\text{pobj}}$ |
| educated_at | ⟨Stephen Hawking, University of Oxford⟩ | $\xleftarrow{\text{nsubj}}$ attended $\xrightarrow{\text{dobj}}$, $\xleftarrow{\text{nsubj}}$ studied $\xrightarrow{\text{prep}}$ at $\xrightarrow{\text{pobj}}$ |
| faculty_of | ⟨Hegel, Heidelberg University⟩ | $\xrightarrow{\text{appos}}$ professor $\xrightarrow{\text{prep}}$ at $\xrightarrow{\text{pobj}}$, $\xrightarrow{\text{compound}}$ professor $\xrightarrow{\text{compound}}$ |

Table 5: Cases where textual contexts correct the predictions. The first relation is the real parent and the second relation is the predicted relation if we only use head and tail entities.

prediction compared to information in the knowledge graph. Table 5 lists several cases where the parent is correctly predicted after using textual contexts. For example, both educated_at and faculty_of look very similar from the perspective of head-tail entities because both of them connect a person to an university. Textual contexts like "attended" and "professor" can reliably distinguish them, improving the performance by a large margin. (2) Dependency paths yield better performance than words in the middle, which indicates that dependency paths can represent the meaning of relations more accurately. (3) Sequential representations outperform bag-of-tokens, indicating that modeling sequences directly is better than only using frequent tokens. Performance numbers of the BiLSTM and CNN are almost the same as word embedding average, indicating that the limited amount of data available may not be enough to train these more complex methods.

## 6.2 Sensitivity Analysis

We use different numbers of instances (10, 100, 200, 500, and 1000) and textual contexts (1, 5, 10, 15, 20) to investigate whether our model is sensitive to these hyperparameters. Figure 5a shows performance of the **BiLSTM with Dep.** model, and we can see that it is relatively stable. More instances can better represent relations, leading to higher accuracy. Since textual contexts are ranked by frequency, incorporating less frequent (thus noisy) contexts slightly hurts the performance.

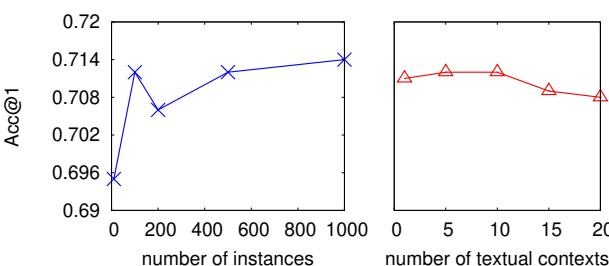

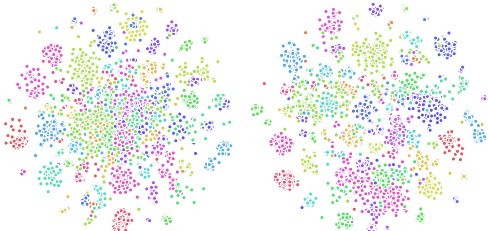

(Figure 5a) Accuracy@1 wrt. different numbers of instances and textual contexts.

(Figure 5b) Visualization (t-SNE) of embeddings before (left) and after (right) MLP. Each point is an instance and is colored according to its parent relation.

### 6.3 Visualization and Analysis

**Relation Entailment ≠ Relation Similarity** Two relations in an entailment relationship are similar to each other to some extent, but entailment means more than being similar, e.g., `screenwriter` and `director` are similar in the sense that both are occupations related to films but there is no entailment between them. This is why selecting the most similar relation from the candidates as the parent can give reasonable performance, but using an additional MLP optimized with entailment annotations can perform much better. We visualize the embeddings before and after the MLP layer in Figure 5b using t-SNE [Maaten and Hinton, 2008], with instances belonging to different parent relations being shown in different colors. It is clear that after the MLP, relations with different parents are better separated, indicating the necessity of using another space for entailment prediction.

**Generalization Requires High-level Abstraction** There are two reasons that make the problem of predicting relation entailment particularly hard: high-level abstraction and data sparsity. Take the first tier in Figure 1 as an example, in order to successfully predict the parent of `designed_by`, models need to capture the

| Parent | Train Rel. | Test Rel. |
|---|---|---|
| `follows` | `has_cause` | `replaces` |
| `instance_of` | `taxon_rank` | `legal_form` |
| `participant` | `performer` | `participating_team` |

Figure 3: Cases where our model fails.

commonality between it and its siblings in the training set (i.e., `author`, `illustrator`, and `founded_by`). This is hard because they are similar on a very abstract level: all of them are special cases of creating. Given the limited number of training samples, discovering this commonality becomes even harder. Figure 3 lists a few cases where our model fails to predict the correct parent (the "Parent" column) for the test relations (the "Test Rel." column), demonstrating these traits.

## 7 Related Work

**Relations Between Relations** Chen et al. [2019] propose a distribution-based method to measure the similarity between relations. Instead of just measuring similarity among relations, we go further to define relation entailment, which can be used to organize relations into a hierarchy, adding value over simple similarity measurement. Zhang et al. [2018] demonstrate the effectiveness of leveraging relation hierarchies in representation learning, grouping relations by similarity into a three-layer hierarchy without considering entailment. Han and Sun [2016] use manually created relation hierarchies to provide more supervision for relation extraction and improve prediction consistency. Some previous works mine entailment relations among textual relational phrases or patterns from a large

text corpus without canonicalization [Lin and Pantel, 2001, Nakashole et al., 2012, Grycner et al., 2015, Kloetzer et al., 2015].

**Relational Knowledge Resources**  Many resources, such as WordNet [Miller, 1995] and Concept-Net [Speer and Havasi, 2013], focus on relationships between words or entities such as synonymy or hyponymy. Relation entailment is similar to hyponymy, but with a specific focus on relations used to connect entities.

## 8  Conclusion

In this paper, we define the task of relation entailment and build a dataset based on Wikidata. Relation entailment prediction has potential applications in many downstream tasks, including representation learning, question answering, relation extraction, and summarization. We establish several baselines using both structured and textual information and provide insights into the task characteristics. Predicting entailment for unseen relations requires high-level abstraction, presenting a unique challenge to learning algorithms. Potential future works include (1) Modeling the relation hierarchy as a structured prediction task to take into account the structure among relations in inference. (2) Extending to textual relations that are by their nature more abundant and diverse.

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
