# OpenReview forum: "Learning Relation Entailment with Structured and Textual Information"
_AKBC.ws/2020/Conference — AKBC 2020_

### Official Review · AnonReviewer2 · 2020-03-24
**Approximating KG entailment via a form of textual entailment**

**Rating:** 7
**Confidence:** 3

**Review:**

This paper addresses the issue of relation entailment, viz., whether a relation r in a knowledge graph entails a relation r', which the authors define as a form of relational containment (r entails r' if r is contained in set theoretical terms by r'). They then propose a data driven methods to sample a gold standard of such containments, that they use to evaluate and/or train unsupervised and supervised relation entailment models.

The authors derive their gold standard from Wikidata, and a number of sampling techniques that are relatively well explained. They rely for their methods on distributed representations of both the relations, and their textual groundings (mapping the triples to bags of words and/or syntactic dependencies derived from Wikipedia snippets via a form of "reverse entity linking" of sorts). They then experiment with on the one hand, similarity functions and on the other hand, CNN and biLSTM encoders. Perhaps unsurprisingly, supervised models perform way better than unsupervised models (from 0.57 to 0.71 accuracy). The models are well described.

This reviewer finds the experiments well described, but still incomplete. Indeed, the authors fail to assess the impact of the different input information modalities (in the input embedding layers of their neural networks) -- triple and word embeddings. Unless the reader is meant to understand that their "base model" in Table 3b) relies only on triple embeddings: this is not clear! Also, this reviewer would like to see results for "text only" models. Is this better than reasoning with the triples or with *both* signals? In similar NLP tasks (think textual entailment), one usually proceeds that way. It would also be interesting, for the sake of completeness, to consider such three cases in Table 3a) (similarity-based approaches). Last, but not least, the scores reported are sometimes quite close. Would it be possible to add the standard deviation of your scores somewhere, in particular for Table 3b) as is common in deep learning literature? This reviewer can't see yet if there was a real improvement or only an statistical fluctuation.

The discussion of the results is quite informative.

---

> ### Author Response · Authors · 2020-04-10
> **Thank you very much for the thorough review!**
>
> (1) This reviewer finds the experiments well described, but still incomplete. Indeed, the authors fail to assess the impact of the different input information modalities (in the input embedding layers of their neural networks) -- triple and word embeddings. Unless the reader is meant to understand that their "base model" in Table 3b) relies only on triple embeddings: this is not clear!
> Thank you for pointing it out. We added a clarification in the caption of Table 3b.
>
> (2) Also, this reviewer would like to see results for "text only" models. Is this better than reasoning with the triples or with *both* signals? In similar NLP tasks (think textual entailment), one usually proceeds that way. It would also be interesting, for the sake of completeness, to consider such three cases in Table 3a) (similarity-based approaches).
> In our preliminary experiments, we observed that only using text features is significantly worse than using structured information (i.e. triples), because text features extracted based on distant supervision assumptions are relatively sparse and noisy. We were not able to re-run experiments with respect to this within the revision period, but we will consider it for the final version.
>
> (3) Last, but not least, the scores reported are sometimes quite close. Would it be possible to add the standard deviation of your scores somewhere, in particular for Table 3b) as is common in deep learning literature? This reviewer can't see yet if there was a real improvement or only an statistical fluctuation.
> Thank you for your suggestion! The differences between different textual information encoding methods are indeed quite small. The major point we want to demonstrate in Table 3b is that textual information can bring large gain over structured information (i.e., the base model). We will aim for either/both of (1) performing multiple runs, and (2) measuring within-run variance through bootstrap tests for the final version.

---

### Official Review · AnonReviewer1 · 2020-03-28
**New dataset introduced along with the task of predicting entailment between relations**

**Rating:** 7
**Confidence:** 4

**Review:**

This paper introduces the task of predicting entailment between canonicalized relations in a knowledge graph. The downstream significance of this work lies in teaching models to understand abstract concepts through predicting entailment between relations, thereby understanding a hierarchy of concepts.

The relations are represented using information from knowledge graphs as well as information extracted from text. A variety of methods are explored for building this representation - KGE methods such as TransE, embedding the context between textual mentions of the relation's entities and distribution based methods.  The prediction task is then formulated as a ranking problem where the correct parent relation should be ranked higher than all others.  The paper is well written and clear except for a few points below.

Comments/Questions::
I feel the nomenclature of unsupervised/supervised scoring functions is a bit misleading. It would be better suited to call the two approaches as non-parametric vs parametric methods.

1. How do the cosine and euclidean similarity metric serve as a scoring function given that they are symmetric ?
2. Is prediction for parent relations done within all the relations only in that tier or all relations?
3. With regards to the relation instance propagation - if the child relations are propagated to the parent, the representation of parent would explicitly include information of the child. I might be missing something but would this not make the task of predicting parent relation trivial since they would be the most similar?

---

> ### Author Response · Authors · 2020-04-10
> **Thank you very much for the thorough review!**
>
> (1) I feel the nomenclature of unsupervised/supervised scoring functions is a bit misleading. It would be better suited to call the two approaches as non-parametric vs parametric methods.
> We argue that the major difference between these two categories is that supervised methods use annotated parent-child relation pairs to optimize the model, while unsupervised methods do not. Since unsupervised methods also involve parameters in knowledge base embeddings, we prefer to stay with unsupervised/supervised categorization.
>
> (2) How do the cosine and euclidean similarity metric serve as a scoring function given that they are symmetric?
> We agree with you that both cosine and euclidean metrics are symmetric. Thus they are not suitable for relation entailment prediction in theory. As explained in the first paragraph of Section 5.1., this is indeed one of our major motivations to develop distribution-based representations and KL-Divergence similarity function, which is asymmetric. However, in experiments we didn’t observe the superiority of asymmetric methods, partially because distribution-based methods are more prone to noise than aggregation-based methods. Devising better asymmetric methods is an interesting avenue for future work.
>
> (3) Is prediction for parent relations done within all the relations only in that tier or all relations?
> As explained in the “Task Definition” Section on page 3, we predict the parent relation among all the parent relations, not only in that tier.
>
> (4) With regards to the relation instance propagation - if the child relations are propagated to the parent, the representation of parent would explicitly include information of the child. I might be missing something but would this not make the task of predicting parent relation trivial since they would be the most similar?
> Sorry for the confusion. Actually when predicting the parent of a particular relation r, we will first remove r’s instances from its parent to avoid making this problem trivial. We added an explanation in footnote 3.

---

### Official Review · AnonReviewer3 · 2020-03-29
**Supervised and Unsupervised Relation Entailment**

**Rating:** 6
**Confidence:** 3

**Review:**

The paper introduces a method for understanding if a relation is direct entailment of another one.  The authors have used the relation already exists in a well-known dataset( Wikidata). I liked the tricks they have used to construct the dataset. (e.g relation sampling, relation expansion, etc.)

The paper in a way that it is a little unclear and hard to follow. For instance, there are continuous mentions of single letter references. Some of the references are explained later ( for example P^r in KL-divergence calculation.)
--It might also help to have a figure showing the general idea of the model and then mentioning that you are running experiments with different settings.
--maybe using the larger equations in separate lines.'
--It might have been interesting to see statistics on relation entailments, such as what percent of the relations have more than children. This might also help with understanding the propagation better.
In might also be interesting to see some qualitative analysis comparing the "TransE", DistMult and ComplEx. Are there domain-dependent. Are there scenarios that the others can outperform TransE?

To sum up, The pros of this paper are as follows:
-Introducing interesting aspects of analysis in the knowledge graph problems.
-analysis of supervised and unsupervised methods to find the entailments in relations.

And weaknesses are:
-The paper is a little hard to follow. Maybe it is better to add a section to define all the repetitive terms. Also adding model figure can help.
-Although I agree that the author has done plenty of experiments, probably some statistic reports on the relations can give more insight into the scope of the problem,

Minor comment: Please high light the highest numbers in the tables.

---

> ### Author Response · Authors · 2020-04-10
> **Thank you very much for the thorough review!**
>
> (1) Some of the references are explained later (for example P^r in KL-divergence calculation.)
> We have modified this equation to make it clear.
>
> (2) It might also help to have a figure showing the general idea of the model and then mentioning that you are running experiments with different settings.
> Thank you for your suggestion! We will consider adding a figure to distinguish different components in our methods (e.g. structured vs textual information) in the final version.
>
> (3) It might have been interesting to see statistics on relation entailments, such as what percent of the relations have more than children. This might also help with understanding the propagation better.
> In our dataset, 86% of parent relations have more than one child. We added this in step 5 in Section 3.
>
> (4) In might also be interesting to see some qualitative analysis comparing the "TransE", DistMult and ComplEx. Are there domain-dependent. Are there scenarios that the others can outperform TransE?
> Thank you for the suggestion! We observed in our experiments that TransE consistently outperforms the other two in different settings, and attribute this to its additive nature, as explained in Section 6.1.
>
> (5) Minor comment: Please highlight the highest numbers in the tables.
> This is now corrected.

---

### Decision · Program_Chairs · 2020-05-01

**Decision:**

Accept

**Comment:**

The paper introduces a method for entailment prediction between relations in a knowledge graph, using the Wikidata dataset. They used a few tricks to construct the dataset (relation sampling, relation expansion, etc.)

Overall, the reviewers agree that this paper deserve publication. However several aspects in the presentation should be improved: notation needs to be made clearer, a figure would help understand the main idea, and statistics on relation entailments would be useful to present.  We strongly recommend authors to take these suggestions into account when preparing the final version.